# The Effect of Enteral Immunonutrition in the Intensive Care Unit: Does It Impact on Outcomes?

**DOI:** 10.3390/nu14091904

**Published:** 2022-05-01

**Authors:** Juan Carlos Lopez-Delgado, Teodoro Grau-Carmona, Javier Trujillano-Cabello, Carlos García-Fuentes, Esther Mor-Marco, Maria Luisa Bordeje-Laguna, Esther Portugal-Rodriguez, Carol Lorencio-Cardenas, Paula Vera-Artazcoz, Laura Macaya-Redin, Juan Francisco Martinez-Carmona, Lidón Mateu-Campos, Maria Gero-Escapa, Rosa Gastaldo-Simeon, Belen Vila-García, José Luis Flordelis-Lasierra, Juan Carlos Montejo-Gonzalez, Lluís Servia-Goixart

**Affiliations:** 1Intensive Care Department, Hospital Universitari de Bellvitge, C/Feixa Llarga s/n, 08907 L’Hospitalet de Llobregat, Spain; 2IDIBELL (Institut d’Investigació Biomèdica Bellvitge, Biomedical Investigation Institute of Bellvitge), Av. de la Gran Via, 199, 08908 L’Hospitalet de Llobregat, Spain; 3Intensive Care Department, Hospital Universitario 12 de Octubre, Av. de Córdoba s/n, 28041 Madrid, Spain; teograu2@gmail.com (T.G.-C.); cgfuentes@salud.madrid.org (C.G.-F.); makalyconru@hotmail.com (J.L.F.-L.); juancarlos.montejo@salud.madrid.org (J.C.M.-G.); 4i+12 (Instituto de Investigación Sanitaria Hospital 12 de Octubre, Research Institute Hospital 12 de Octubre), Av. de Córdoba s/n, 28041 Madrid, Spain; 5Intensive Care Department, Hospital Universitari Arnau de Vilanova, Av. Alcalde Rovira Roure, 80, 25198 Lleida, Spain; jtruji9@gmail.com (J.T.-C.); lserviag@gmail.com (L.S.-G.); 6IRBLLeida (Institut de Recerca Biomèdica de Lleida Fundació Doctor PiFarré, Lleida Biomedical Research Institute’s Doctor PiFarré Foundation), Av. Alcalde Rovira Roure, 80, 25198 Lleida, Spain; 7Intensive Care Department, Hospital Universitario Germans Trias i Pujol, Carretera de Canyet, s/n, 08916 Badalona, Spain; esthermormarco@gmail.com (E.M.-M.); luisabordeje@gmail.com (M.L.B.-L.); 8Intensive Care Department, Hospital Clínico Universitario de Valladolid, Av. Ramón y Cajal, 3, 47003 Valladolid, Spain; esther_burgos@hotmail.com; 9Intensive Care Department, Hospital Universitari Josep Trueta, Av. de França, s/n, 17007 Girona, Spain; carol_lorencio@hotmail.com; 10Intensive Care Department, Hospital de la Santa Creu i Sant Pau, C/Sant Quintí, 89, 08041 Barcelona, Spain; pvera@santpau.cat; 11Intensive Care Department, Complejo Hospitalario de Navarra, C/Irunlarrea, E, 31008 Pamplona, Spain; lauramacaya@hotmail.com; 12Intensive Care Department, Hospital Regional Universitario Carlos Haya, Av. de Carlos Haya, 84, 29010 Málaga, Spain; jf.mtnez88@gmail.com; 13Intensive Care Department, Hospital General Universitario de Castellón, Avda. de Benicàssim, 128, 12004 Castelló de la Plana, Spain; mateuli@hotmail.com; 14Intensive Care Department, Hospital Universitario de Burgos, Av. Islas Baleares, 3, 09006 Burgos, Spain; mariagero.esc@gmail.com; 15Intensive Care Department, Hospital de Manacor, Carretera Manacor-Alcudia, s/n, 07500 Manacor, Spain; rousgastaldi@hotmail.com; 16Intensive Care Department, Hospital Universitario Infanta Cristina, Av. 9 de Junio, 2, 28981 Parla, Spain; belenvilag@yahoo.es

**Keywords:** enteral nutrition, immunonutrition, intensive care unit, protein delivery, outcomes, inflammatory response

## Abstract

Background: The present research aimed to evaluate the effect on outcomes of immunonutrition (IMN) enteral formulas during the intensive care unit (ICU) stay. Methods: A multicenter prospective observational study was performed. Patient characteristics, disease severity, nutritional status, type of nutritional therapy and outcomes, and laboratory parameters were collected in a database. Statistical differences were analyzed according to the administration of IMN or other types of enteral formulas. Results: In total, 406 patients were included in the analysis, of whom 15.02% (61) received IMN. Univariate analysis showed that patients treated with IMN formulas received higher mean caloric and protein intake, and better 28-day survival (85.2% vs. 73.3%; *p* = 0.014. Unadjusted Hazard Ratio (HR): 0.15; 95% CI (Confidence Interval): 0.06–0.36; *p* < 0.001). Once adjusted for confounding factors, multivariate analysis showed a lower need for vasopressor support (OR: 0.49; 95% CI: 0.26–0.91; *p* = 0.023) and continuous renal replacement therapies (OR: 0.13; 95% CI: 0.01–0.65; *p* = 0.049) in those patients who received IMN formulas, independently of the severity of the disease. IMN use was also associated with higher protein intake during the administration of nutritional therapy (OR: 6.23; 95% CI: 2.59–15.54; *p* < 0.001), regardless of the type of patient. No differences were found in the laboratory parameters, except for a trend toward lower triglyceride levels (HR: 0.97; 95% CI: 0.95–0.99; *p* = 0.045). Conclusion: The use of IMN formulas may be associated with better outcomes (i.e., lower need for vasopressors and continuous renal replacement), together with a trend toward higher protein enteral delivery during the ICU stay. These findings may ultimately be related to their modulating effect on the inflammatory response in the critically ill. NCT Registry: 03634943.

## 1. Introduction

Malnutrition remains highly prevalent (i.e., as high as 50% depending on the severity and type of patient) in the intensive care unit (ICU), and the deterioration of the nutritional status is strongly associated with worse outcomes [1]. Worsening nutritional status is related to metabolic alterations (i.e., hypermetabolic state) and the degree of inflammatory response due to surgical and trauma injury or infection that patients suffer during critical illness [2]. These alterations worsen the nutritional status and impact on outcomes in clinical practice, leading to higher episodes of sepsis, alterations in wound healing, higher severity of myopathy and weaning failure, among other complications, unless appropriate nutritional therapy is performed [3].

The use of specific enriched enteral nutrition (EN) formulas with immunonutrients may modulate the function of the immune system and the inflammatory response, which could improve outcomes [4]. The use of enteral immunonutrition (IMN) formulas may be beneficial to the trauma and surgical patient, especially during the perioperative period, whereas its use in medical patients remains controversial [5,6]. Despite a theoretical rationale for their use and positive trends toward a reduction in infection complications, IMN has failed to demonstrate a clinical benefit in recent contemporary trials [7,8,9,10]. However, the formulas used in these trials varied widely in their composition (i.e., selenium, glutamine, and antioxidants) and dosage compared with the currently commercialized immunomodulatory formulas (i.e., arginine, ω-3 polyunsaturated fatty acids (PUFAs) and nucleotides) [11,12]. There is no full agreement as to dosage and use of the different immunonutrients for specific populations of patients because the available formulas vary in their composition and dosage, and iatrogenic underfeeding in some patients may contribute to the lack of effect of the immunonutrients due to inappropriate dosage [5].

The present study aimed to evaluate the outcomes with the use of IMN formulas enriched with arginine, PUFAs, and nucleotides compared with other available enteral formulas among a heterogeneous population of critical care patients. We also separately analyzed the influence of IMN on the different types of patients admitted to the ICU (i.e., medical, trauma, surgical patients), the impact of IMN on caloric and protein delivery and, finally, its influence on metabolic and nutrition-related laboratory parameters.

## 2. Materials and Methods

A multicenter prospective observational study was conducted in 37 Spanish ICUs between April 2018 and July 2018. All consecutive adult patients (i.e., >18 years old) requiring artificial nutritional support with an expected ICU stay >72 h were included in the analysis. Patients admitted to the ICU for postoperative recovery and ICU monitoring, without needing specific therapy for organ support, were excluded from the study. Only patients who required EN exclusively were included for the present research, and those requiring exclusive or complementary parenteral nutrition were excluded.

The study was approved by a central Institutional Ethics Committee (Comité d’Ètica i Assajos Clínics de Hospital Universitari de Bellvitge; Barcelona, Spain) with approval number PR401/17. Informed consent was waived due to the observational nature of our study according to Spanish law. Patients included in the present study were obtained from the Evaluation of Nutritional Practices in the Critical Care registry (ENPIC Study; ClinicalTrials.gov Identifier: NCT03634943). Despite not making any attempt to standardize or influence the general ICU care and nutritional approach of these patients due to the observational nature of this study, EN dosage was usually prescribed in around 25–35% of cases on day 1, 60–70% on day 2, and full dose on day 3, based on current practice. Indeed, in order to evaluate the eligibility of the hospitals, participants were asked to provide information regarding their clinical nutrition practices and the degree of adherence to the current guidelines (e.g., presence of a nutritional protocol, healthcare giver involved in artificial nutritional support) [13].

Data were prospectively extracted from the medical registry of each patient and collected in a local database for analysis purposes. The REDCap^®^ electronic data capture tools hosted by the data coordinating center from the Catalan Institute of Health at the Hospital Arnau de Vilanova (Lleida, Spain) were used for this purpose. Basic demographic and clinical data (i.e., diagnosis, type of admission and comorbidities), together with nutritional assessment (Subjective Global Assessment (SGA) and modified Nutrition Risk in the Critically Ill (mNUTRIC) score) and ICU prognosis scores (Acute Physiology and Chronic Health Evaluation (APACHE) II, Simplified Acute Physiology Score (SAPS) II and Sequential Organ Failure Assessment (SOFA)) were collected. Nutritional practices such as EN delivery, type of formula, the amount of energy and protein intake, EN-related complications (i.e., residual gastric volume, diarrhea, vomiting, aspiration and mesenteric ischemia) and outcomes (i.e., need for vasopressor support during the ICU stay, renal replacement therapies (RRT), mechanical ventilation, respiratory tract and catheter-related infections) until ICU discharge or for a maximum of 14 days were recorded. ICU and hospital mortality were followed up for 28 days. EN-related complications were defined based on recently established guidelines [14]. Non-nutritional calories (i.e., dextrose infusion and Propofol) and enteral protein supplementation were also considered for the mean energy and protein intake calculations. Required energy and protein intakes were those calculated by the physician in charge.

Blood samples for laboratory analysis were collected via a central venous catheter at the ICU. These samples were drawn from the patients on admission, day 3, day 7, and at discharge to measure metabolic and nutrition-related parameters (i.e., albumin, prealbumin, total protein, leukocytes, lymphocytes, C-reactive protein, and the different lipid profiles, such as total cholesterol, high-density lipoprotein, low-density lipoprotein, and triglyceride levels). The measurements were performed according to international standard laboratory procedures. Before the study began, the investigators checked that each local laboratory had specific accreditation for medical laboratories adhering to the international organization for standardization (ISO).

The IMN enteral formula used (Vegenat Healthcare^®^, Badajoz, Spain) contained 1.51 Kcal·mL^−1^ (17.3 g of carbohydrates·100 mL^−1^; 46% of the total energy supplied), high protein content (8.3 g·100 mL^−1^; 22% of the total energy supplied) and lipids (5 g·100 mL^−1^; 30% of the total energy supplied). The composition of the protein content was 50% casein, 25% whey protein, and 25% vegetal protein (i.e., peas). This formula is enriched with arginine (1 g·100 mL^−1^), nucleotides (200 mg·100 mL^−1^), and also PUFAs (1.6 g·100 mL^−1^; 399 mg·100 mL^−1^ of eicosapentaenoic and docosahexaenoic fatty acids).

The data entered were cleaned from August to November 2018 to identify errors, inconsistencies, and omissions in order to provide optimal data completeness. Data queries were sent back to the participating investigators for verification, and a second check was performed and the database was closed in May 2020. We compared the differences among the subgroups of patients receiving IMN enteral formula and those receiving other types of EN formulas.

### Statistical Analyses

Statistical analysis was conducted using PASW statistics 20.0 (SPSS Inc., Chicago, IL, USA). Data were expressed as frequencies and percentages, means and standard deviations, or median and interquartile range, or both, when appropriate. We analyzed differences between patients receiving IMN enteral formula and those receiving other types of EN formulas. Trauma and surgical patients were analyzed together for statistical purposes (i.e., insufficient number of patients in each subgroup) when we analyzed subgroups (i.e., type of patients). The Mann–Whitney U-test or, when appropriate, the two-sample *t*-test for comparisons between groups was used. The χ²-test was used to evaluate categorical prognostic factors in order to identify differences among subgroups.

Multivariate analysis was carried out using a backward stepwise logistic regression, and an adjusted multiple stepwise Cox regression analysis was performed when needed (i.e., to add time perspective), to identify factors associated with the use of IMN. Variables with *p* < 0.1 were included in the initial model and according to the investigators’ criteria. Change-in-estimate criterion and backwards deletion with a 10% cutoff were used to eliminate confounding variables from our final models. We tested for interactions between the variables that we introduced into the multivariate analyses to avoid destabilization of the different analyses. We performed adjustment for age, body mass index, SGA, NUTRIC score, ICU scores, and significant differences before admission between the compared cohort of patients in order to avoid any influence of the nutritional status, nutritional risk, and severity of illness when analyzing the outcomes. The duration of nutritional therapy was also included in all analyses to add a time perspective.

In all cases, the Kolmogorov–Smirnov test and D’Agostino–Pearson omnibus normality test were used to check the normal distribution of our population and to assess the goodness-of-fit of the final regression models. Survival analysis was carried out using the Kaplan–Meier estimator for the two subgroups. A two-tailed *p* value <0.05 was considered statistically significant.

## 3. Results

### 3.1. Population Included in the Study

During the study period, 406 patients who received EN exclusively were included for analysis (Figure 1). Baseline characteristics, which include nutritional assessment, mean amount of delivered caloric and protein intake, EN-related complications, and outcomes of the patients admitted to the ICU for the whole cohort are described in Table 1. Regarding the type of subpopulations included in the study, 71.2% (289) of the patients were medical patients, 15% (61) were trauma patients, and 13.8% (56) were surgical patients.

### 3.2. Univariate Analysis and Nutrition Delivery

We classified the patients according to whether they received an IMN formula during EN administration or not (Table 2). Patients who received IMN were younger, had a lower incidence of Chronic Obstructive Pulmonary Disease, represented a higher percentage of trauma and surgical patients, had better nutritional risk (e.g., lower mNUTRIC score), and also presented a lower degree of organ failure (e.g., lower SOFA score on admission). The mean caloric and protein delivery were both higher in those receiving IMN during the nutritional therapy, which was more pronounced during the first week of nutrition therapy, especially regarding protein delivery. In fact, we found similar rates of EN-related complications, or even higher rates in the IMN subgroup; however, we found better caloric and protein delivery in the patients who received IMN formula, especially for the administration of nutritional therapy during the first two weeks of the ICU stay (Figure 2 and Table 3). The mean volume of EN delivered was similar between groups (845 ± 384 mL in the EN subgroup vs. 892 ± 342 mL in the EN-INM subgroup; *p = 0.85*) (see Appendix A). The same applied for the mean ratio of delivered/required energy (0.79 ± 0.6 in the EN subgroup vs. 0.84 ± 0.5 in the EN-INM subgroup; *p = 0.45*) and delivered/required protein (0.77 ± 0.6 in the EN subgroup vs. 0.85 ± 0.6 in the EN-INM subgroup; *p = 0.55*), although it differed slightly at some time-points (see Appendix A). We found similar mean carbohydrate delivery in the EN-INM subgroup (149 ± 63.2 g·day^−1^ in the EN subgroup vs. 152.4 ± 58.9 g·day^−1^ in the EN-INM subgroup; *p = 0.65*), except for a slight trend toward better delivery during the first days of EN (see Appendix A), whereas no difference was shown regarding the mean amount of delivered lipids (41.7 ± 17.1 g·day^−1^ in the EN subgroup vs. 43.8 ± 17.1 g·day^−1^ in the EN-INM subgroup; *p = 0.82*) (see Appendix A).

Arginine and PUFAs are the main pharmacologically active components of the IMN enteral formula [5,6]. The mean daily dose of arginine in the IMN group was 8.9 ± 4.5 g·day^−1^ during the entire study period (i.e., during the entire EN administration or at least for the first 14 days). Trauma and surgical patients received a dose of 9.6 ± 4.1 g·day^−1^ whereas medical patients received 8.1 ± 4.6 g·day^−1^. This mean daily dose of arginine was higher from day four until the end of EN (i.e., when patients theoretically received full EN) in all patients who received IMN (11.9 ± 3.8 g·day^−1^). The same applied for subgroups: trauma and surgical patients (12.1 ± 3.5 g·day^−1^) and medical patients (11.3 ± 4.3 g·day^−1^) received a higher arginine dose during this period. Patients who were fed other types of EN formulas received a minimal mean dose of arginine: 1.3 ± 1.9 g·day^−1^. Regarding PUFAs, the mean daily dose in the IMN group was 12.9 ± 5.3 g·day^−1^ during the entire study period. Trauma and surgical patients received a dose of 13.1 ± 4.1 g·day^−1^, whereas medical patients received 11.8 ± 5.6 g·day^−1^. This mean daily dose of PUFAs was higher from day four until the end of EN in all the patients who received IMN (15.9 ± 5.8 g·day^−1^). The same applied for subgroups: trauma and surgical patients (17.1 ± 6.5 g·day^−1^), and medical patients (16.1 ± 5.9 g·day^−1^) received a higher PUFAs dose. Finally, the mean daily dose of nucleotides that the EN-INM subgroup received was 1752 ± 687 mg.

### 3.3. Outcome Results and Multivariate Analysis

Regarding outcomes, a lower need for vasopressor support and RRT during the ICU stay, together with a trend toward lower mortality was also seen with IMN administration. Survival analysis revealed that patients who received IMN formula showed a trend toward better 28-day survival (Figure 3). This trend was confirmed (Hazards Ratio (HR): 0.15; 95% Confidence Interval (CI): 0.06–0.36; *p* < 0.001), once adjusted for confounding factors (i.e., age, comorbidities, severity of disease, type of patient, ICU scores, nutritional risk scores, and duration of nutritional therapy).

Multivariate analysis, also adjusted for confounding factors, revealed differences among patients who received IMN in terms of organ support: they required vasopressor support and RRT less frequently during their ICU stay. We also showed a higher amount of mean protein delivery with IMN formula (Table 4).

### 3.4. Subgroup Analysis by Type of Patients and Laboratory Results

We also analyzed the differences between patients receiving IMN enteral formula or not among different subpopulations based on the type of ICU patients (i.e., trauma, surgical, and medical). The subgroup of surgical and trauma patients who received IMN showed lower mortality, whereas medical patients required vasopressor support and RRT less frequently during their ICU stay (Table 5). Adjusted multivariate analysis showed that all the subpopulations received a higher mean amount of protein delivery during nutritional therapy (Table 6).

Finally, we found no significant differences in laboratory parameters (Table 7) except for a trend toward lower triglyceride blood levels (HR: 0.97; 95% CI: 0.95–0.99; *p = 0.045*) during the ICU stay.

## 4. Discussion

The results of the present research may suggest a potential clinical benefit with the use of IMN formula in terms of outcomes and survival in ICU patients who have to be fed by the enteral route. Patients receiving IMN experienced a reduction in the need for vasopressor therapy and RRT during their ICU stay. In addition, we also found a trend toward better caloric and protein delivery. All these potential benefits may be explained by a more appropriate inflammatory response thanks to the immunomodulatory effect that this type of enteral formula may exert [12].

ICU patients suffer from an intense inflammatory response caused by surgical stress, trauma injury, and medical conditions, such as sterile (e.g., hemorrhagic shock) or infectious disease, that often results in vasoplegia and clinically significant hypotension [15,16]. Under these circumstances, vasopressors are required to maintain appropriate mean arterial pressure and blood flow through organs and tissues, allowing adequate nutrient and substrate transportation, which ultimately may prevent organ dysfunction [15,17]. The components of the IMN enteral formula, such as arginine, PUFAs, and nucleotides, are involved in multiple metabolic processes associated with immune function [5,6]. Arginine is involved in the correct functioning of the immune system, connective tissue repair, and nitric oxide, which is a signaling molecule that participates in the production of reactive oxygen species involved in the inflammatory response [18]. PUFAs have potent anti-inflammatory properties with the suppression of proinflammatory transcription factors due to their modulation of eicosanoid production [19,20]. It has been hypothesized that arginine and PUFAs may have a synergistic effect, which is capable of restoring lymphocyte function and interleukin production [18].

Acute kidney injury (AKI) represents a major complication in the ICU and is associated with worst outcomes and higher risk of mortality, especially when patients require RRT [21]. The occurrence of AKI is explained by the interplay of several factors, which include inflammation and oxidative stress, microvascular dysfunction, and the adaptive response of the tubular epithelial cell [22]. Avoiding hypotension and guaranteeing adequate blood perfusion are among the factors that must be considered for AKI recovery and to prevent its progression [23]. In consequence, the lower vasopressor and RRT needs seen in patients under IMN may be linked to the modulator effect that this type of nutritional therapy exerts over postinjury inflammation. These findings were not detected within the different subpopulations, which was probably due to an insufficient sample size.

As highlighted above, an inappropriate nutritional therapy with lower calorie and protein intake is associated with worse outcomes in ICU patients [1]. Immune dysfunction and inflammation are both enhanced in the presence of malnutrition and vice versa, and nutritional therapy per se may help avoid and prevent this phenomenon [24]. We found a higher protein delivery in patients fed with IMN formula without any influence on better performance (i.e., higher mean volume of delivered EN and ratio delivered/required energy and protein), even when those patients were analyzed by subpopulation. In recent observational studies, higher protein intake was associated with better outcomes in the ICU [25]. Thus, the better outcomes presented in our results may be explained by better tolerance and a more appropriate protein composition of the IMN formula (e.g., higher protein content and density), and not just due to the influence of IMN on the inflammatory response. In addition, high-protein intake (especially early) may result in a reduction of the negative protein balance and muscle loss caused by the illness.

Despite positive clinical results, we were unable to find any effect on current laboratory markers of inflammation, nutrition, and immunity [26]. Only a slight improvement in the lipid profile was seen in our patients, with the reduction of triglyceride blood levels. However, lipid metabolism is related to the metabolic response to critical illnesses: hypertriglyceridemia may reflect adipose tissue lipolysis and persistent catabolism [27]. In addition, lipid metabolism and gut microbiota are both affected by dietary lipids, which may also help regulate metabolism [28].

Our study has several limitations, which are mainly related to the observational nature of the study and the heterogeneity of the patients. We were unable to evaluate in depth the different components of IMN enteral formulas, and even the effects produced by these immunonutrients may be limited by other factors, such as the better protein provision they received during nutritional therapy. The multicenter nature, the significant number of patients within our population, and the nutritional therapy follow-up until day 14 after ICU admission are among the strengths of this study. The sample size could be adequate for the whole ICU population, but it may be not optimal for subgroup analysis based on the low number of surgical and trauma patients. Thus, the results from subgroup analysis, especially the trauma and surgical subgroup, should be considered with caution. In order to avoid the confounding influence of illness severity, nutritional status, and population heterogeneity, we accounted for potential confounders along with the finest statistical performance that aimed to minimize their influence (described in the Methods section). Even though the results are debatable, they are clinically relevant in terms of outcomes during the ICU stay.

## 5. Conclusions

In summary, the use of IMN formulas may be associated with better outcomes (i.e., lower need for vasopressors and continuous RRT), together with a trend toward better survival, and a higher protein enteral delivery during the ICU stay. This may ultimately be related to their modulating effect on the inflammatory response in the critically ill.

## Figures and Tables

**Figure 1 nutrients-14-01904-f001:**
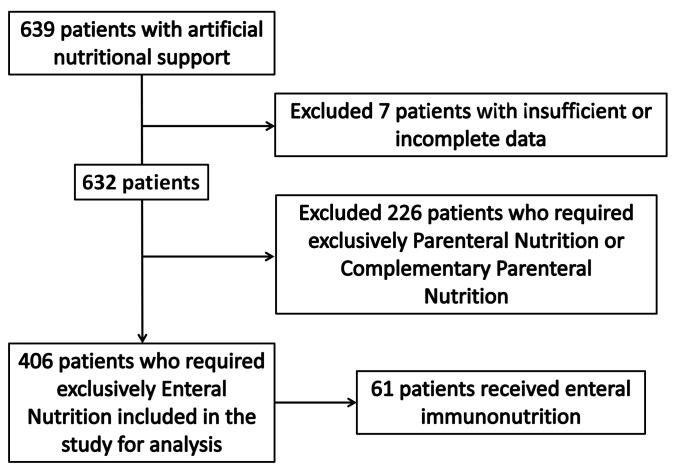
Study flow of patients included in the study.

**Figure 2 nutrients-14-01904-f002:**
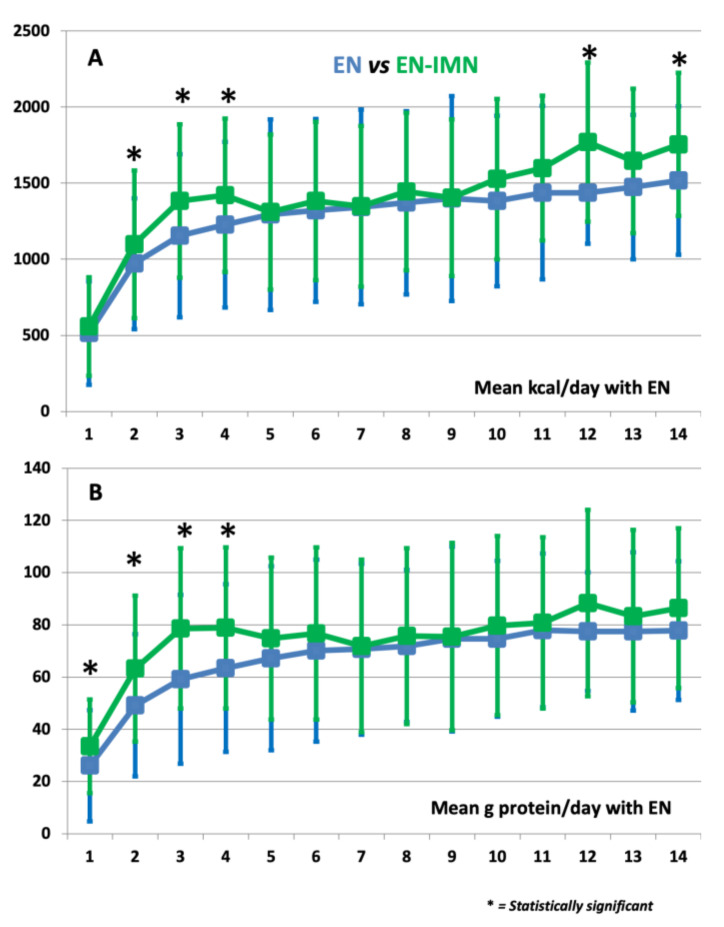
Differences in mean caloric (**A**) and protein (**B**) delivery during nutritional therapy in the ICU of patients receiving standard or immunonutrition enteral formula.

**Figure 3 nutrients-14-01904-f003:**
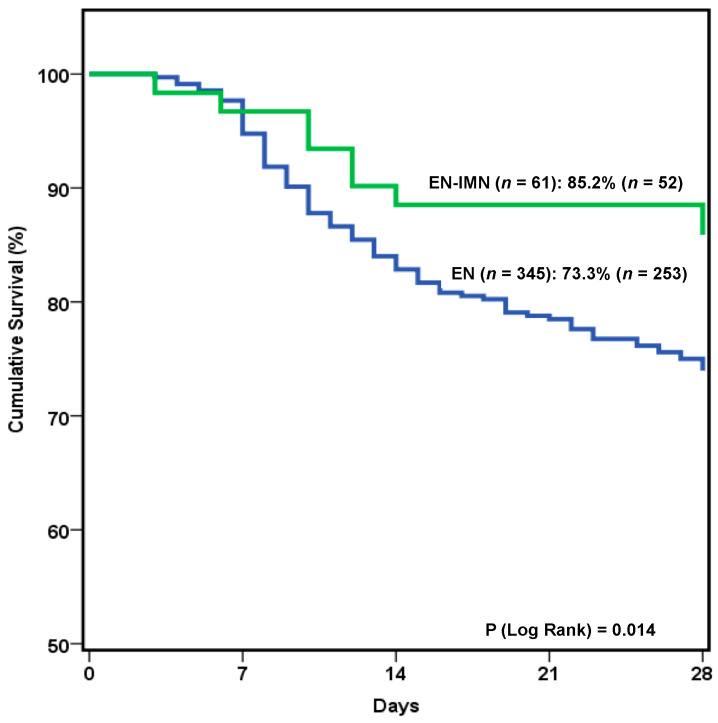
Kaplan–Meier curves and log rank test showing 28-day survival according to patients receiving immunonutrition or other types of enteral formulas.

**Table 1 nutrients-14-01904-t001:** Characteristics, nutritional therapy, and outcomes of patients receiving enteral nutrition.

Baseline Characteristics and Comorbidities		
Mean age (years)		60.8 ± 15
Sex (male)		67.4% (274)
BMI (Kg·m^−2^)		28.2 ± 6.3
Alcohol		12.81% (52)
Diabetes		25.37% (103)
Hypertension		42.36% (172)
COPD		17.98% (73)
AMI		15.02% (61)
Chronic Liver Disease		5.42% (22)
Chronic Renal Failure		10.34% (42)
Immunosuppression		10.34% (42)
Neoplasia		15.27% (62)
Type of patient	Medical	71.18% (289)
	Trauma	15.02% (61)
	Surgery	13.79% (56)
APACHE II		20 (15–25)
SAPS II		48.35 ± 17.39
SOFA (on admission)		7.07 ± 3.2
Malnutrition (based on SGA)		34.41% (139)
mNUTRIC Score		3.97 ± 2.15
Characteristics of nutritional support		
Patients with IMN		15.02% (61)
Early enteral nutrition (<48 h)		76.8% (310)
Mean EN administration (days)		8.5 (4–17)
Mean Kcal/Kg/day *		15.4 ± 5.2
Mean g protein/Kg/day *		0.75 ± 0.34
EN-related complications	↑ GRV	11.4% (46)
	Diarrhea	8.6% (35)
	Vomiting	1.2% (5)
	Aspiration	0
	Mesenteric ischemia	0.74% (3)
Outcomes		
Mechanical ventilation		97.54% (396)
Days on mechanical ventilation		13.2 ± 13.8
Vasopressor support		73.4% (298)
Days on vasopressor support		3.26 ± 3.53
RRT needs		10.1% (41)
Respiratory tract infection		25.12% (102)
Catheter-related infections		6.4% (26)
Mean ICU stay (days)		13 (8–22)
Mean hospital stay (days)		25 (16–42)
28-day mortality		24.9% (101)

EN: Enteral Nutrition; IMN: Immunonutrition Formula; BMI: Body Mass Index; COPD: Chronic Obstructive Pulmonary Disease; AMI: Acute Myocardial Infarction; APACHE II: Acute Physiology and Chronic Health Disease Classification System II; SAPS: Simplified Acute Physiology Score; SOFA: Sequential Organ Failure Assessment; SGA: Subjective Global Assessment; mNUTRIC: modified Nutrition Risk in the Critically Ill; ↑ GRV: Elevated Gastric Residual Volume; ICU: Intensive Care Unit; RRT: Renal Replacement Therapy. * During the entire administration of EN or at least for the first 14 days.

**Table 2 nutrients-14-01904-t002:** Differences among patients receiving immunonutrition formulas and other types of enteral formulas.

		EN *n* = 345 (84.98%)	EN-IMN *n* = 61 (15.02%)	*p*-Value
Baseline characteristics and comorbidities				
Mean age (years)		61.45 ± 15.13	56.7 ± 16.35	**0.03**
Sex (male)		64.93% (224)	81.97% (50)	**0.01**
BMI (Kg·m^−2^)		27.24 (24.23–31.15)	26.24 (23.44–29.33)	0.07
Alcohol		13.04% (45)	11.48% (7)	0.89
Diabetes		25.8% (89)	22.95% (14)	0.75
Hypertension		43.77% (151)	34.43% (21)	0.22
COPD		20.29% (70)	4.92% (3)	**0.003**
AMI		14.78% (51)	16.39% (10)	0.89
Chronic Liver Disease		5.51% (19)	4.92% (3)	0.99
Chronic Renal Failure		10.14% (35)	11.48% (7)	0.93
Immunosuppression		11.01% (38)	6.56% (4)	0.36
Neoplasia		14.49% (50)	19.67% (12)	0.33
Type of patient	Medical	75.94% (262)	44.26% (27)	**<0.001**
	Trauma	11.59% (40)	34.43% (21)	
	Surgery	12.46% (43)	21.31% (13)	
APACHE II		20 (15–25)	18 (13–23)	0.06
SAPS II		49.01 ± 17.43	44.12 ± 16.7	0.06
SOFA (on admission)		7.21 ± 3.24	6.26 ± 2.87	**0.03**
Malnutrition (based on SGA)		35.76% (123)	26.67% (16)	0.22
mNUTRIC Score		4.08 ± 2.16	3.35 ± 2.05	**0.02**
Characteristics of nutritional support				
Early enteral nutrition (<48 h)		73.9% (255)	90.1% (55)	0.15
Mean of EN administration (days)		8 (4–17)	9 (4–18)	0.557
Mean Kcal/Kg/day *		14.4 ± 5.69	16.24 ± 5.31	**0.01**
Mean g protein/Kg/day *		0.74 ± 0.34	0.9 ± 0.31	**<0.001**
EN-related complications	↑ GRV	11.30% (39)	11.47% (7)	0.89
	Diarrhea	7.83% (27)	13.11% (8)	0.21
	Vomiting	1.16% (4)	0.2% (1)	0.56
	Aspiration	0	0	NA
	Mesenteric ischemia	0.87% (3)	0% (0)	0.99
Outcomes				
Mechanical ventilation		97.68% (337)	96.72% (59)	0.65
Days on mechanical ventilation		13.39 ± 13.88	12.38 ± 13.96	0.73
Vasopressor support		75.65% (261)	60.66% (37)	**0.02**
Days on vasopressor support		3.27 ± 3.48	3.21 ± 3.85	0.45
RRT needs		11.59% (40)	1.64% (1)	**0.03**
Respiratory tract infection		26.09% (90)	19.67% (12)	0.36
Catheter-related infections		6.67% (23)	4.92% (3)	0.78
Mean ICU stay (days)		14 (9–23)	14 (8–21)	0.69
Mean hospital stay (days)		25 (16–41.5)	27.5 (16–47.5)	0.29
28-day mortality		26.67% (92)	14.75% (9)	0.07

EN: Enteral Nutrition; IMN: Immunonutrition Formula; BMI: Body Mass Index; COPD: Chronic Obstructive Pulmonary Disease; AMI: Acute Myocardial Infarction; APACHE II: Acute Physiology and Chronic Health disease Classification System II; SAPS: Simplified Acute Physiology Score; SOFA: Sequential Organ Failure Assessment; SGA: Subjective Global Assessment; mNUTRIC: modified Nutrition Risk in the Critically Ill; ↑ GRV: Elevated Gastric Residual Volume; ICU: Intensive Care Unit; RRT: Renal Replacement Therapy, NA: Not applicable. *p*-Values that are statistically significant are written in bold. * During the entire administration of EN or at least for the first 14 days.

**Table 3 nutrients-14-01904-t003:** Mean caloric and protein delivery during enteral nutrition therapy.

Day	Mean kcal/day			Mean g Protein/day		
	EN	EN-IMN	*p*	EN	EN-IMN	*p*
1	515 ± 324	557 ± 341	0.35	26.1 ± 17.9	33.5 ± 21.3	**0.01**
2	971 ± 485	1099 ± 430	**0.04**	49.2 ± 27.9	63.2 ± 27.3	**<0.001**
3	1155 ± 503	1383 ± 536	**0.003**	59.1 ± 30.7	78.6 ± 32.3	**<0.001**
4	1228 ± 504	1420 ± 544	**0.02**	63.4 ± 30.9	78.8 ± 32.1	**0.002**
5	1293 ± 509	1310 ± 625	0.85	67.2 ± 31	74.8 ± 35.2	0.13
6	1322 ± 520	1382 ± 601	0.54	70.1 ± 32.9	76.7 ± 34.8	0.24
7	1344 ± 527	1348 ± 639	0.97	70.7 ± 33.1	71.9 ± 32.7	0.83
8	1371 ± 518	1445 ± 602	0.42	71.9 ± 33.7	75.7 ± 29.1	0.51
9	1399 ± 515	1404 ± 673	0.96	74.6 ± 35.8	75.5 ± 35.4	0.88
10	1383 ± 526	1527 ± 560	0.15	74.7 ± 34.3	79.7 ± 29.8	0.39
11	1438 ± 476	1599 ± 569	0.10	77.9 ± 32.8	80.8 ± 29.4	0.63
12	1437 ± 523	1770 ± 336	**0.003**	77.4 ± 35.7	88.3 ± 22.7	0.05
13	1475 ± 475	1647 ± 474	0.12	77.5 ± 33	83.3 ± 30.3	0.43
14	1517 ± 471	1755 ± 487	**0.07**	77.8 ± 30.6	86.4 ± 26.5	0.27

EN: Enteral Nutrition Formula; IMN: Immunonutrition Formula. *p*-Values that are statistically significant are written in bold.

**Table 4 nutrients-14-01904-t004:** Multivariate analysis—variables associated with the use of immunonutrition.

**Dependent Variable—Use of IMN**	**Odds Ratio** **(95% Confidence Interval)**	* **p** * **-Value**
Type of patient (trauma)	1.490 (0.630–3.640)	0.37
Mean protein delivery (g·Kg^−1^·day^−1^) *	6.230 (2.590–15.541)	**<0.001**
Vasopressor support	0.440 (0.230–0.845)	**0.012**
RRT Needs	0.432 (0.231–0.850)	**0.049**

RRT: Renal Replacement Therapy. *p*-Values that are statistically significant are written in bold. * During the entire administration of EN or at least for the first 14 days.

**Table 5 nutrients-14-01904-t005:** Differences among patients receiving immunonutrition formulas and other types of enteral formulas in trauma and surgical (**A**) and medical (**B**) patients.

**A**		**EN** *n* **= 83 (70.94%)**	**EN-IMN** ***n* = 34 (29.06%)**	* **p** * **-Value**
Baseline characteristics and comorbidities				
Mean age (years)		59.59 ± 18.2	53.09 ± 17.53	0.061
Sex (male)		72.29% (60)	94.12% (32)	**0.012**
BMI (Kg·m^−2^)		27.29 (25.21–30.55)	25.91 (23.44–27.78)	**0.036**
Alcohol		10.84% (9)	8.82% (3)	0.95
Diabetes		19.28% (16)	20.59% (7)	0.99
Hypertension		43.37% (36)	20.59% (7)	**0.035**
COPD		13.25% (11)	0	**0.032**
AMI		10.84% (9)	20.59% (7)	0.27
Chronic Liver Disease		0	2.94% (1)	0.29
Chronic Renal Failure		9.64% (8)	11.76% (4)	0.89
Immunosuppression		1.2% (1)	5.88% (2)	0.20
Neoplasia		13.25% (11)	17.65% (6)	0.57
APACHE II		18 (13–23.5)	17.5 (12–22.75)	0.83
SAPS II		45.85 ± 16.51	43 ± 19.15	0.29
SOFA (on admission)		7.25 ± 3.08	6.35 ± 2.88	0.13
Malnutrition (based on SGA)		22.89% (19)	18.18% (6)	0.80
mNUTRIC Score		3.55 ± 2.26	3.12 ± 2.19	0.37
Characteristics of nutritional support				
Early enteral nutrition (<48 h)		78.3% (65)	91% (31)	0.26
Mean of EN administration (days)		8 (3.5–17.5)	9.5 (4–17.75)	0.38
Mean Kcal/Kg/day *		13.12 ± 5.74	17.09 ± 4.06	**<0.001**
Mean g protein/Kg/day *		0.66 ± 0.33	0.97 ± 0.26	**<0.001**
EN-related complications	↑ GRV	9.64% (8)	11.7% (4)	0.45
	Diarrhea	7.23% (6)	17.65% (6)	0.10
	Vomiting	1.2% (1)	0	0.99
	Aspiration	0	0	NA
	Mesenteric ischemia	0	0	NA
Outcomes				
Mechanical ventilation		100% (83)	100% (34)	NA
Days on mechanical ventilation		13.35 ± 11.56	10.79 ± 6.45	0.63
Vasopressor support		77.1% (64)	67.6% (23)	0.16
Days on vasopressor support		4.79 ± 4.18	3.04 ± 3.29	0.23
RRT needs		9.6% (8)	2.9% (1)	0.28
Respiratory tract infection		22.9% (19)	17.6% (6)	0.62
Catheter-related infections		7.2% (6)	2.9% (1)	0.67
Mean ICU stay (days)		14 (8–22.5)	14.5 (8–21)	0.70
Mean hospital stay (days)		26 (16.5–38)	32 (21.75–45.25)	0.21
28-day mortality		19.28% (16)	2.94% (1)	**0.022**
**B**		**EN** ***n* = 262 (90.66%)**	**EN-IMN** ***n* = 27 (9.34%)**	** *p* ** **-Value**
Baseline characteristics and comorbidities				
Mean age (years)		62.04 ± 14.01	61.26 ± 13.73	0.68
Sex (male)		62.6% (164)	66.67% (18)	0.83
BMI (Kg·m^−2^)		27.2 (24.22–31.55)	26.35 (23.96–31.03)	0.69
Alcohol		13.74% (36)	14.81% (4)	0.77
Diabetes		27.86% (73)	25.93% (7)	0.97
Hypertension		43.89% (115)	51.85% (14)	0.54
COPD		22.52% (59)	11.11% (3)	0.22
AMI		16.03% (42)	11.11% (3)	0.78
Chronic Liver Disease		7.25% (19)	7.41% (2)	0.98
Chronic Renal Failure		10.31% (27)	11.11% (3)	**0.08**
Immunosuppression		14.12% (37)	7.41% (2)	0.55
Neoplasia		14.89% (39)	22.22% (6)	0.39
APACHE II		21 (16–26)	19 (14–23.5)	0.22
SAPS II		50.02 ± 17.63	45.38 ± 13.73	0.34
SOFA (on admission)		7.19 ± 3.29	6.15 ± 2.92	**0.09**
Malnutrition (based on SGA)		39.85% (104)	37.04% (10)	0.93
mNUTRIC Score		4.25 ± 2.1	3.63 ± 1.86	0.12
Characteristics of nutritional support				
Early enteral nutrition (<48 h)		72.5% (190)	88.80% (24)	0.28
Mean of EN administration (days)		8 (4–17.75)	9 (3.5–17.5)	0.90
Mean Kcal/Kg/day *		13.81 ± 4.62	15.16 ± 6.48	0.52
Mean g protein/Kg/day *		0.71 ± 0.32	0.83 ± 0.39	0.10
EN-related complications	↑ GRV	11.83% (31)	11.11% (3)	0.87
	Diarrhea	8.02% (21)	7.41% (2)	0.98
	Vomiting	1.15% (3)	3.7% (1)	0.32
	Aspiration	0	0	NA
	Mesenteric ischemia	1.15% (3)	0	0.99
Outcomes				
Mechanical ventilation		96.95% (254)	92.59% (25)	0.23
Days on mechanical ventilation		13.4 ± 14.58	14.62 ± 20.36	0.76
Vasopressor support		75.2% (197)	51.8% (14)	**0.018**
Days on vasopressor support		3.35 ± 3.54	2.48 ± 3.32	0.11
RRT needs		12.2% (32)	0	**0.04**
Respiratory tract infection		27.1% (71)	22.22% (6)	0.65
Catheter-related infections		6.49% (17)	7.41% (2)	0.69
Mean ICU stay (days)		13 (9–23)	13 (9–19.5)	0.86
Mean hospital stay (days)		25 (15–42)	22.5 (12.75–53.5)	0.93
28-day mortality		29.01% (76)	29.6% (8)	0.98

EN: Enteral Nutrition Formula; IMN: Immunonutrition Formula; BMI: Body Mass Index; COPD: Chronic Obstructive Pulmonary Disease; AMI: Acute Myocardial Infarction; APACHE II: Acute Physiology and Chronic Health disease Classification System II; SAPS: Simplified Acute Physiology Score; SOFA: Sequential Organ Failure Assessment; SGA: Subjective Global Assessment; mNUTRIC: modified Nutrition Risk in the Critically Ill; ↑ GRV: Elevated Gastric Residual Volume; ICU: Intensive Care Unit; RRT: Renal Replacement Therapy; NA: Not applicable. *p*-Values that are statistically significant are written in bold. * During the entire administration of EN or at least for the first 14 days.

**Table 6 nutrients-14-01904-t006:** Multivariate analysis—variables associated with the use of immunonutrition in trauma and surgical and medical patients.

Dependent Variable—Use of IMN	Odds Ratio (95% Confidence Interval)	*p*-Value
Trauma and Surgical		
Mean protein delivery (g·Kg^−1^·day^−1^) *	5.711 (2.876–6.892)	**0.02**
Vasopressor support	0.850 (0.641–1.278)	0.19
In-hospital mortality	1.012 (0.929–1.129)	0.09
Medical		
Mean protein delivery (g·Kg^−1^·day^−1^) *	4.630 (3.041–6.801)	**0.001**
Vasopressor support	0.687 (0.520–1.181)	0.11
RRT needs	0.905 (0.850–1.620)	0.35

RRT: Renal Replacement Therapy. *p*-Values that are statistically significant are written in bold. * During the entire administration of EN or at least for the first 14 days.

**Table 7 nutrients-14-01904-t007:** Laboratory markers of the different subgroups measured during nutritional therapy.

		EN	EN-IMN	*p*-Value
Serum protein				
Albumin (mg·dL^−1^)	Day 1	30.37 ± 6.08	33.57 ± 7.18	**0.002**
	Day 3	28.17 ± 5.39	29.12 ± 5.70	0.35
	Day 7	27.78 ± 5.83	28.04 ± 5.35	0.81
	ICU discharge	29.88 ± 6.09	29.73 ± 5.66	0.88
	Δ (day 1-ICU discharge)	−3.85 ± 5.83	−4.48 ± 6.17	**0.001**
Prealbumin (mg·L^−1^)	Day 1	153.28 ± 81.05	178.77 ± 50.14	**0.04**
	Day 3	150.43 ± 80.31	150.28 ± 56.90	0.99
	Day 7	213.14 ± 110.06	201.78 ± 92.87	0.68
	ICU discharge	216.28 ± 92.82	207.64 ± 102.97	0.75
	Δ (day 1-ICU discharge)	59.47 ± 101.54	31.27 ± 97.55	0.38
Total Protein (g·dL^−1^)	Day 1	5.72 ± 0.87	5.96 ± 0.96	0.12
	Day 3	5.29 ± 0.75	5.47 ± 0.81	0.25
	Day 7	5.37 ± 0.76	5.35 ± 0.91	0.91
	ICU discharge	5.81 ± 0.97	6.13 ± 0.81	**0.04**
	Δ (day 1-ICU discharge)	0.29 ± 1.12	0.78 ± 0.92	0.27
Lipid profile				
Total Cholesterol (mg·dL^−1^)	Day 1	132 ± 39	136 ± 41	0.58
	Day 3	128 ± 42	132 ± 32	0.63
	Day 7	143 ± 41	127 ± 36	0.16
	ICU discharge	156 ± 49	144 ± 45	0.35
	Δ (day 1-ICU discharge)	21 ± 52	7.9 ± 30	0.17
HDL (mg·dL^−1^)	Day 1	38.1 ± 18.9	38.1 ± 15.1	0.98
	Day 3	60.9 ± 26.2	77.1 ± 13.6	0.50
	Day 7	33.9 ± 19.7	25.9 ± 11.2	0.18
	ICU discharge	34.03 ± 17.2	31.5 ± 7	0.59
	Δ (day 1-ICU discharge)	−2.8 ± 6.8	1.5 ± 8.9	0.26
LDL (mg·dL^−1^)	Day 1	65.8 ± 31.4	75.9 ± 32.9	0.16
	Day 3	158.9 ± 25.7	246.1 ± 43.7	0.37
	Day 7	84.3 ± 49.4	69.4 ± 25.1	0.35
	ICU discharge	91.4 ± 39.6	87.7 ± 44.3	0.75
	Δ (day 1-ICU discharge)	25.7 ± 13.5	2.5 ± 23.5	**0.03**
Triglycerides (mg·dL^−1^)	Day 1	135 ± 73	128 ± 69	0.61
	Day 3	156 ± 99	133 ± 47	**0.08**
	Day 7	168 ± 104	160 ± 107	0.77
	ICU discharge	160 ± 89	136 ± 64	0.20
	Δ (day 1-ICU discharge)	26 ± 106	−10 ± 39	**0.02**
Inflammatory markers				
CRP (mg·L^−1^)	Day 1	26.65 ± 45.89	54.82 ± 76.97	**0.07**
	Day 3	77.34 ± 106.35	91.55 ± 148.97	0.54
	Day 7	95.51 ± 120.77	67.96 ± 125.95	0.17
	ICU discharge	58.11 ± 77.22	77.30 ± 62.61	0.25
	Δ (day 1-ICU discharge)	−64.41 ± 140.70	−17.12 ± 91.08	0.11
Leukocytes (×10^9^·L^−1^)	Day 1	14.69 ± 6.27	13.84 ± 5.29	0.18
	Day 3	10.61 ± 5.28	9.54 ± 4.46	**0.04**
	Day 7	8.8 ± 6.12	10.1 ± 5.65	0.18
	ICU discharge	11.15 ± 5.59	9.72 ± 3.26	0.11
	Δ (day 1-ICU discharge)	−2.5 ± 7.3	−2.7 ± 5.5	0.84
Lymphocytes (×10^9^·L^−1^)	Day 1	1.77 ± 1.42	1.69 ± 0.92	0.58
	Day 3	1.59 ± 1.66	1.09 ± 0.76	0.37
	Day 7	0.92 ± 0.6	1.22 ± 0.84	0.30
	ICU discharge	1.56 ± 1.1	1.4 ± 0.65	0.15
	Δ (day 1-ICU discharge)	0.096 ± 1.49	−0.003 ± 1.10	0.63

EN: Enteral Nutrition Formula; EN-IMN: Immunonutrition Formula; HDL: High-Density Lipoprotein; LDL: Low-Density Lipoprotein; CRP: C-Reactive protein; Δ: delta. *p*-Values that are statistically significant are written in bold.

## Data Availability

The data presented in this study are available previous contact to the corresponding author of the present manuscript. Data requests should be evaluated by local ethics committee in order to be agree with legal requirements.

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
