# Peer review of "The Effect of Enteral Immunonutrition in the Intensive Care Unit: Does It Impact on Outcomes?"

_nutrients, 2022, doi:10.3390/nu14091904_

Round 1

Reviewer 1 Report

This prospective multicenter study evaluates the effect of enteral immunonutrition on ICU morbidity and mortality.  To date, studies on immunonutrition in critically ill yield contradicting results and current guidelines do not recommend its use on a routine basis. 

The study by Lopez-Delgado adds knowledge in this field, however, there are some points that need to be adressed 

1) Did you perform a sample size calculation before starting enrollment? When comparing subgroups (surgical / trauma / medical) sample size per group gets very small. It is questionable whether the impact on mortality etc. is reliable with such small groups.

2) Did any of  the study participants receive additional parenteral nutrition?

3) How was enteral nutrition initiated in your patients? At full prescribed dosage / 50% / 25%? How fast was the dosage increased? 

4) average caloric intake was quite low (15 kcal/kg/day) in the study population, however patients receiving immunonutrition achieved higher caloric intake compared to the standard group. This difference was even more pronounced when comparing medical vs. surgical / trauma patients. The positive effects could be mainly attributed to the higher caloric intake.

The same holds true for protein intake.

5) Patients in the immunonutrition group were younger, healthier, less prone to undernutrion and were more likely to have trauma or surgery as reason for ICU admission. This could sway the results towards better outcome. however, multivariate analysis confirmed the findings even after adjusting for confounders.

Minor comments:

line 121: NUTRIIC score instead of NUTRIC score

line 307: feed should say "fed"

Reviewer 2 Report

This study was conducted to evaluate the effect on outcomes of immunonutrition(IMN) enteral formulas during ICU stay. And authors concluded that the use of IMN formulas may be associated with better outcomes.

I read this manuscript interestingly.

Despite of your good study and result, I think several points should be improved.

  1. What the differences delivered EN  (in EN vs. EN-IMN)

         - Calories, protein, lipid and special nutrients (Can you describe as table or appendix?)

     2. How many EN was delivered? 

        If delivered amounts were different between both group, the result might be affected. And the ratio of delivered energy and protein /required energy and protein should be added. 

     3. I think the EN group was more serous (eg. SOFA, APACHE II, SAPS, mNutric, Vasopressor, etc). So, if the matched analysis is added, the result can be more improved.   

   4. In surgical patients, mortality was different in both group. How do you think this result, and did you suspect this result is affected by IMN?

     - I think due to small number of surgical patients, the results might be confused, despite of the difference of severity (however, the p value was not significant in severity)

Reviewer 3 Report

1. There is no clear and conclusive evidence that the “immunonutrition group” shows better results and lower mortality because they received immunonutrion because they had better conditions from the beginning:

1.1 The immunonutrition group had younger age and less severe comorbidities (COPD.RTT).

1.2 The immunutrition group was younger (p  0.03).

2. The immunutrition group was supplemented with a higher dose of protein (p<0.001) and a higher dose of energy (p 0.01).

For these reasons, it is necessary to formulate the importance of immunonutrion much more carefully in the discussion and in the conclusion.

3. I also have the following questions:

3.1 What type of plant protein has been used in enteral nutrition.

3.2 What was the average daily dose of Arginine in the immunonutrient in preparation.

3.3 What was the average daily dose of PUFA (especially eicosapentaenoic and docosahexaenoic fatty acids).

Conclusion:

This is an interesting and useful study, but needs to complete the required data and modify the formulations.

Round 2

Reviewer 1 Report

My comments and questions were sufficiently addressed by the authors. I have no further comments.

Author Response

The language of the present version of the manuscript has been reviewed by native speakers (Helena Kruyer from the Language Services of the Institute of Biomedical Research of Bellvitge)

Reviewer 2 Report

Thank you for yours good answer. 

The mean of first question is the component of EN formula.

As your document, IMN formula has higher calories (1.5 kcal/ml) and more protein incluing immune enhancing nutrients. 

So, I think if you add the comparison of components (CHO (calories), protein, lipid, etc. ) (not all, but major differences).

Author Response

See document attached
